# Anesthetic Management for a Pregnant Patient with Bilateral Vocal Cord Granuloma Using High-Flow Nasal Cannula Oxygenation with Oxygen Reserve Index Monitoring: A Case Report

**Hyo Sung Kim [1], Seok Kyeong Oh [1,*] , Jae Eun Lee [1], Hyun Ah Lee [1] and Jae Gu Cho [2]**

[1] Department of Anesthesiology and Pain Medicine, Korea University Guro Hospital, Korea University College of Medicine, Seoul 08308, Republic of Korea; haha60606@gmail.com (H.S.K.); kellyont@gmail.com (J.E.L.); christinalee@gmail.com (H.A.L.)

[2] Department of Otolaryngology-Head and Neck Surgery, Korea University Guro Hospital, Korea University College of Medicine, Seoul 08308, Republic of Korea; jgcho@korea.ac.kr

* Correspondence: nanprayboy@korea.ac.kr; Tel.: +82-2-2626-1437

**Abstract:** Anesthetic management for pregnant patients suffering from airway pathology poses unique challenges. The presence of a bilateral vocal cord granuloma adds further complexity to anesthetic management as it can potentially cause a compromised airway and respiratory distress. This case presents a pregnant patient with a bilateral vocal cord granuloma who underwent anesthesia using high-flow nasal cannula (HFNC) oxygenation and oxygen reserve index (ORi) monitoring. A 33-year-old pregnant woman, who underwent intubation six months ago, experienced hoarseness and was ultimately diagnosed with a bilateral granuloma. Due to the significant airway obstruction, neither intubation nor ventilation was feasible, thereby requiring a surgical intervention. Before the surgical removal, the patient's oxygenation was ensured using HFNC oxygenation. After confirming the sufficient oxygenation of the patient with an ORi of 0.38, the operation commenced, and as it lasted approximately 3 min, the patient was able to tolerate the brief period without additional oxygen supply. Post-surgical excision, mask bagging, and HFNC oxygenation was resumed, driving the ORi to 0.39; then, the operation was resumed. Throughout the procedure, the $SpO_2$ remained above 98. The combination of HFNC and ORi ensured adequate oxygenation and allowed for the early detection of hypoxemia during the procedure. This approach may be a good option for managing granulomas.

**Keywords:** intubation granuloma; airway management; general anesthesia; pregnancy

## 1. Introduction

Anesthetic management for pregnant patients suffering from airway pathology poses unique challenges. A pregnancy-induced reduction in the functional residual capacity, in conjunction with an increased oxygen consumption, lowers the oxygen reserve of the mother. Also, engorged vessels and edema of the upper airway during pregnancy increase the risk of upper airway obstruction and trauma [1–3]. Furthermore, deoxygenation in pregnant patients can have tragic consequences due to its impact on the fetus.

Vocal cord granulomas are benign lesions that are commonly found in the posterior glottis. They can result from various causes, such as gastroesophageal reflux, intubation trauma, and vocal abuse. Granulomas caused by intubation usually appear 2 to 10 weeks after the event. Although vocal cord granulomas can occur in individuals of any age or gender, they are predominantly found in women who have undergone intubation [4–6]. Symptoms typically include hoarseness, a sensation of a lump or discomfort, dyspnea, coughing, and hemoptysis [7]. The primary treatment approach is conservative, but the course of treatment varies based on the underlying cause. Surgery is indicated in

cases of airway obstruction and histopathologic diagnosis [8,9]. In this particular case, a pregnant patient had a symptomatic bilateral vocal cord granuloma. Due to the symptoms and limitations in the medication options for the fetus, surgery under general anesthesia was chosen as the optimal option. However, the presence of a giant bilateral vocal cord granuloma poses challenges during general anesthesia, as it could interfere with intubation and ventilation, increasing the risk of hypoxia, especially considering its potential impact on the fetus.

Preoxygenation prior to endotracheal intubation is crucial, as there is an unavoidable period of apnea during the insertion of the endotracheal tube. Adequate preoxygenation in patients without specific underlying conditions prevents desaturation for up to 9 min [10]. To minimize apnea duration during intubation and enhance oxygen levels, the use of high-flow nasal cannula (HFNC) during intubation may be considered [11]. HFNC delivers heated and humidified oxygen at a maximum flow of 60 L/min without the need for invasive techniques. Additionally, HFNC generates positive airway pressure, effectively clearing the nasopharyngeal dead space [12].

Pulse oximetry is a vital tool for the straightforward monitoring of patient oxygenation. However, pulse oximetry cannot accurately measure oxygen content that is fully saturated above 97%. On the other hand, multiwavelength pulse co-oximetry, specifically oxygen reserve index (ORi) monitoring, allows for the measurement of the $PaO_2$ within the range of 100 to 200 mmHg [13]. ORi monitoring is particularly useful during preoxygenation, extending the acceptable apnea time and identifying inadequate preoxygenation by detecting interruptions in oxygen delivery [13].

This case report highlights the airway management of a patient with a giant vocal cord granuloma undergoing surgery, with a specific emphasis on the patient's oxygen reserve. To ensure sufficient oxygen levels and enhance safety, the anesthesia plan involved the use of HFNC oxygenation combined with ORi monitoring, representing a novel approach for managing anesthesia in patients with potential difficult intubation.

## 2. Case Report

Written informed consent was obtained from the patient for the publication of this case report.

We report a case of a 33-year-old pregnant woman (height, 153 cm; weight, 61 kg) at 13 weeks' gestation. She had been experiencing hoarseness and a bilateral granuloma, requiring surgical intervention.

Six months ago, she was admitted to Korea University Guro Hospital's emergency room with an unstable mental condition. She experienced severe vomiting and diarrhea prior to admission, and shortly after the urgent admission, she suffered a cardiac arrest with pulseless electrical activity. Cardiac compressions were initiated right away, and she was intubated with a tube with a 7.5 inner diameter. After three cycles of cardiopulmonary resuscitation, spontaneous circulation was restored. Computed tomography revealed a hemoperitoneum that was caused by an ovarian cyst rupture. Her vital signs and hypovolemic condition were managed in the emergency room. Her mental conditions improved with the recovery of her vital signs. However, four hours after intubation, she accidently removed the intubated tube. Shortly after this incident, the patient was transferred to the operating room for emergency laparoscopic exploration. The intubation was reperformed before the surgery in the operating room. The surgery lasted for one hour. After the surgery, she was transferred to the intensive care unit, and her endotracheal tube was maintained due to the possibility of unstable vital signs. Fifteen hours after the surgery, extubation was performed, and she was able to breathe spontaneously without discomfort. After spending two days in the ICU, she recovered well and resumed her daily life.

Approximately three months later, she became pregnant. At 11 weeks' gestation, she returned to Korea University Guro Hospital's emergency room with complaints of hoarseness and dyspnea. During a medical examination, she mentioned that mild hoarseness had persisted since her last admission, but it worsened three days prior, and was accompanied

by dyspnea and a sore throat. She was referred to the otolaryngology department, where an examination of her glottis with indirect laryngoscopy revealed bilateral large vocal process granulomas that almost completely obstructed her glottis (Figure 1a). Initially, 20 mg of oral prednisolone daily for 7 days, following 10 mg prednisolone daily for 7 days with a proton pump inhibitor were prescribed for treatment. Despite the limited information, evidence suggests that proton pump inhibitors are not teratogenic [14]. However, as a steroid classified as category C of teratogen and due to its potential of an increased risk of cleft palate during the first trimester, a short period of steroid usage was recommended by the obstetrician [15]. Her symptoms improved with oral medication. However, upon discontinuation of the medications after one week, her symptoms recurred. Due to the progression in the size of the granuloma and the burden of continuing steroid treatment, surgical intervention was deemed necessary. (Figure 1b).

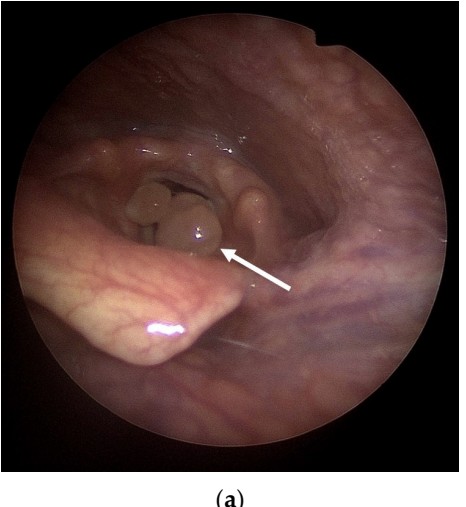 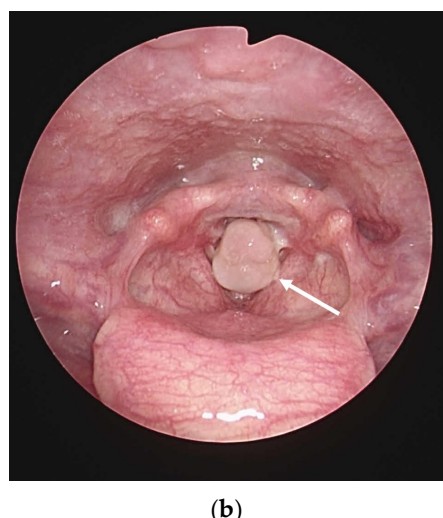

(**a**)            (**b**)

**Figure 1.** (**a**) Indirect laryngoscope image of vocal cord process granuloma of the patient when she was first admitted at the otolaryngology department and (**b**) 1 month after discharge (1 week after the discontinuation of oral steroids). Arrow indicates vocal cord granuloma.

Due to the severe airway obstruction, intubation and ventilation were not deemed feasible. The medical team, including the anesthesiologist and otolaryngologist, discussed potential surgical options in advance. Tracheostomy was considered but was deemed excessive for the patient's condition. Jet ventilation was considered as an option, but the healthcare providers in our center did not frequently practice it. Considering the anticipated short duration of the surgical procedure, the anesthetic plan focused on preoxygenation and maintaining adequate oxygen reserves to sustain any potential apneic periods. HFNC oxygenation was chosen to ensure sufficient oxygen supply before anesthesia, and ORi monitoring was utilized as a tool to predict the oxygen reserve and decreases in oxygen saturation during general anesthesia.

The essential laboratory data, including CBC, electrolytes, and blood coagulation, were within normal ranges. Due to dyspnea, the patient underwent a chest X-ray with shielding to reduce fetal radiation [16] and a pulmonary function test. The chest X-ray showed no abnormalities, but the pulmonary function test revealed an extrathoracic obstruction pattern, causing difficulty with inhalation. Due to her history of cardiac arrest, she underwent echocardiography. The echocardiography revealed a grade 1 impairment of the left ventricle diastolic function and normal left ventricle systolic function with an ejection fraction of 60–65%. A pre-anesthetic physical examination was conducted. She had grade 2 Mallampati, and she had unrestricted neck extension with a full range of motion. Although she experienced hoarseness and dyspnea, there were no apparent neck or mandible swelling issues that could hinder mask bagging.

In the operating room, standard monitoring was implemented, including pulse oximetry, non-invasive blood pressure, and electrocardiography. A bispectral index (BIS<sup>TM</sup>, Medtronic, Minneapolis, MN, USA) was used to estimate the anesthetic depth, and electromyography (EMG, TwitchView®, Blink, Seattle, WA, USA) was used to evaluate the degree of muscle relaxation. In addition, the ORi was continuously monitored through pulse oximetry using a sensor attached to the patient's finger (Radical-7™, Masimo Corp., Irvine, CA, USA), allowing for an ongoing assessment of the oxygen reserve. Following the placement of monitoring, the pulse oximetry displayed a stable $SpO_2$ of 100% along with consistent vital signs.

Prior to the surgical mass removal, the patient received sufficient oxygen through HFNC oxygenation for five minutes. HFNC was then used continuously until the bilateral granuloma excision, at which point mask bagging became possible. After confirming the patient's oxygen status through an ORi reading of 0.35, total intravenous anesthesia with 2% propofol and remifentanil was initiated with the target-controlled infusion of propofol (Ce, 3–4 ng/mL) and remifentanil (Ce, 3–4 ng/kg), targeting the BIS under 60. A loss of consciousness was induced, but mask bagging was impeded by the mass. Twenty milligrams of rocuronium was immediately administered to facilitate laryngeal microscopic surgery and prevent movement, at which point the ORi was 0.32. After confirming muscle relaxation using train of four count (TOF) monitoring with a count of 0, the operation commenced. During the surgery, the anesthesia was maintained with a combination of propofol and remifentanil, targeting a BIS value under 60. The larynx was exposed with suspension laryngoscope after the patient laid in the Boyce position. The bilateral granuloma was successfully removed with cup-head forceps and scissors. The surgical views of the granuloma before excision and after the excision were captured during the surgery (Figure 2). At that moment, the ORi decreased to 0 and the $SpO_2$ was 96%, and the surgical procedure was put on hold for a minute. Mask ventilation was performed without difficulty, and adequate mask bagging followed by HFNC oxygenation restored the ORi to 0.38 and restored the $SpO_2$ to 99% before the surgery was completed (Figure 3). At the end of the procedure, anesthetic agents were discontinued, and the patient regained self-respiration after receiving 120 mg of sugammadex. After confirming stable vital signs and spontaneous respiration, she was transferred to the post-anesthetic care unit (PACU). She did not experience complications in the PACU. On post-operative day (POD) 1, her follow-up indirect laryngoscope showed the absence of a granuloma on the vocal cords (Figure 4). Also, she did not have symptoms of dyspnea. She was discharged on POD 1 since she did not have any complications after the surgery.

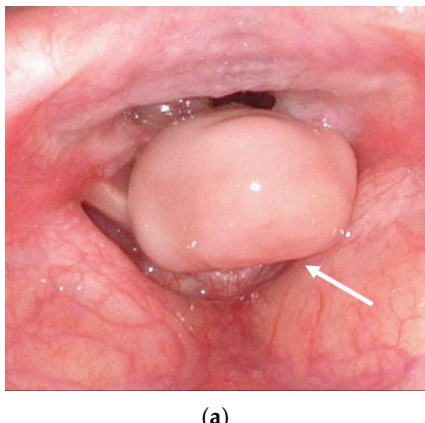 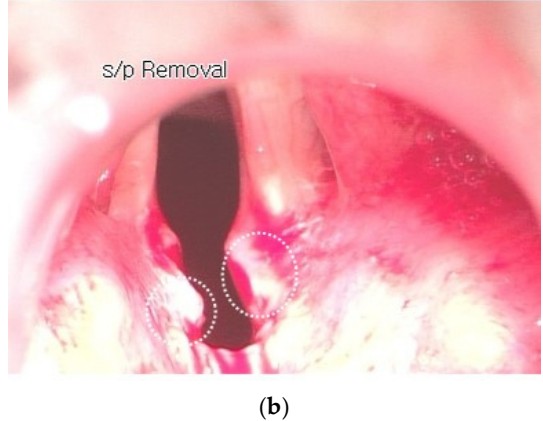

(**a**)   (**b**)

**Figure 2.** (**a**) The surgical view of granuloma before excision and (**b**) after vocal cord granuloma excision. Arrow indicates vocal cord granuloma, and dotted circles indicate excision site of granuloma. s/p: status post.

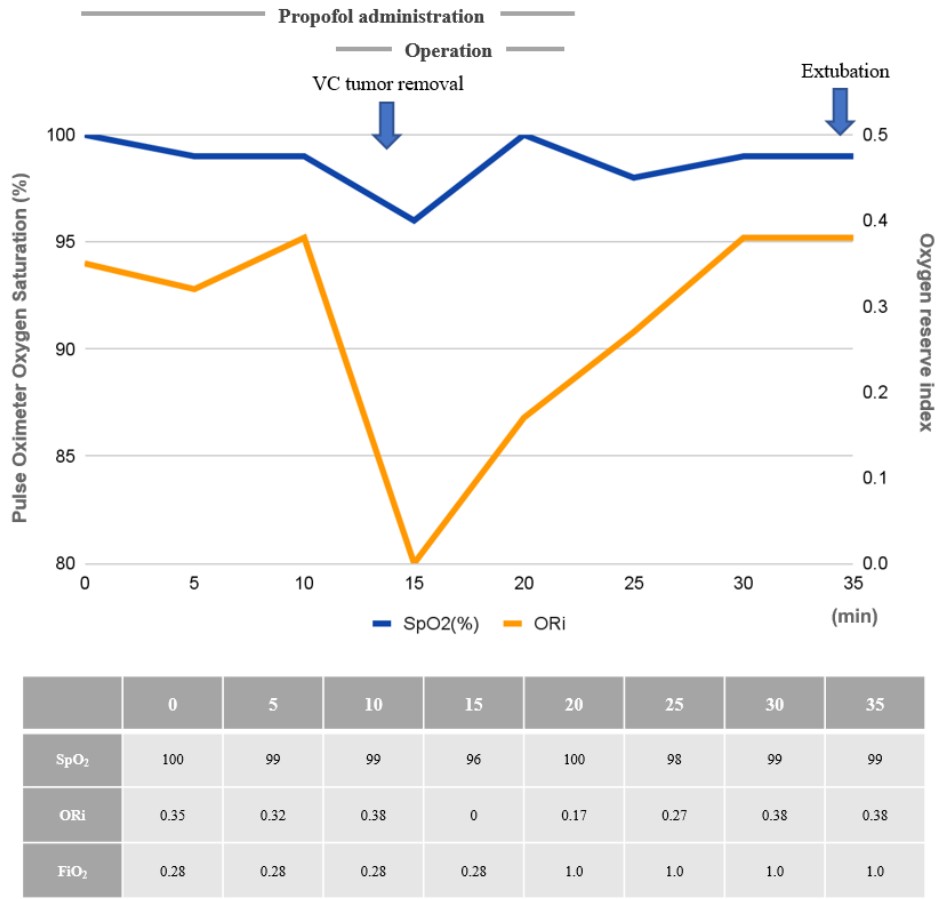

**Figure 3.** Change in pulse oximeter oxygen saturation (SpO$_2$) and oxygen reserve index (ORi).

|                 | 0    | 5    | 10   | 15   | 20   | 25   | 30   | 35   |
|-----------------|------|------|------|------|------|------|------|------|
| SpO$_2$         | 100  | 99   | 99   | 96   | 100  | 98   | 99   | 99   |
| ORi             | 0.35 | 0.32 | 0.38 | 0    | 0.17 | 0.27 | 0.38 | 0.38 |
| FiO$_2$         | 0.28 | 0.28 | 0.28 | 0.28 | 1.0  | 1.0  | 1.0  | 1.0  |

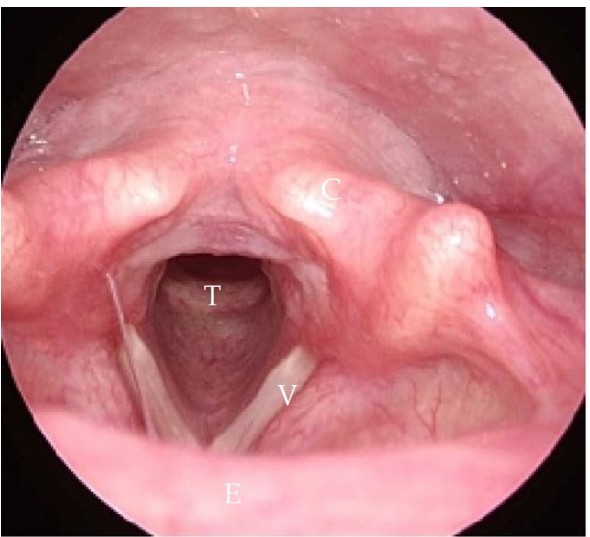

**Figure 4.** Indirect laryngoscope image after the surgery. C: cartilage, E: epiglottis, T: trachea, V: vocal cord.

Due to the patient's increased risk of bleeding resulting from partial placental previa, a condition in which the placenta partially covers the cervix, a decision was made to schedule a cesarean section. This surgical procedure, aimed at ensuring the safety of both the mother and the baby, was scheduled to take place 5 months after the vocal surgery. She underwent an indirect laryngoscope prior to the surgery, revealing clear vocal cords without any granulomas. In addition to the clear findings on the indirect laryngoscope, the patient showed an absence of any recurring symptoms that existed prior to the development of

the bilateral granuloma. Specifically, she did not complain of dyspnea, hoarseness, or any discomfort in her throat. These reassuring reports further indicated the positive outcome of the procedure and her overall improved vocal health. To ensure a comprehensive evaluation, an otolaryngologic consultation was conducted to assess the condition of her vocal cords. During the consultation, the clinician carefully examined the patient's vocal cord movement and did not observe any abnormal or irregular patterns. This further confirmed the successful outcome of the surgery and provided additional reassurance regarding the patient's vocal cord health. As a result, it was deemed feasible for her to undergo general anesthesia with endotracheal intubation. Despite undergoing surgery under spinal anesthesia, she did not experience any complications, and the neonate did not show signs of distress perioperatively.

## 3. Discussion

Intubation-related trauma can cause a vocal process granuloma [17]. The occurrence rate of intubated granulomas after endotracheal intubation is around 0.1~3.3% [18]. While most cases are resolved spontaneously, some cases require surgical intervention [19]. In this case, the patient underwent repeated intubations of prolonged duration, potentially leading to vocal process abrasion and granuloma formation. Females have a higher frequency of reported cases due to the relatively larger size of the tube used in comparison to the larynx, and the thinner mucosal layer of the arytenoid vocal cord (59 μm in females compared to 97 μm in males) [20]. Moreover, physiological changes during pregnancy could affect gastroesophageal reflux due to the increased abdominal pressure and congestion of capillaries, and can contribute to edema in the respiratory mucosa of various structures, including the nasopharynx, pharynx, larynx, arytenoid, and posterior cricoid cartilage, the trachea, etc. In this case, the patient presented with a bilateral granuloma occupying a significant portion of the vocal cord, highlighting the invasive nature of endotracheal intubation and the importance of careful attention. Furthermore, to minimize complications associated with endotracheal intubation, it is crucial to select an appropriate tube size to prevent the abrasion of the vocal process. An excessive manipulation of the intubated tube can also lead to vocal cord injury, necessitating proper fixation and reduced movement of the tube.

Treatment options for vocal process granulomas include voice therapy, managing the gastroesophageal reflux, and various medical interventions, such as surgery, antibiotics, steroids, observation, irradiation, botulinum neurotoxin injection, and membranous vocal fold augmentation [5]. Surgery is typically reserved for cases involving airway obstruction or suspicion of carcinoma due to the high recurrence rates associated with granulomas. In this case, the patient's giant granuloma and persistent dyspnea necessitated surgical management, considering the risks to the well-being of the fetus. This case report highlights the successful airway management of the patient despite the challenges posed by the narrowed glottis opening, demonstrating the effective use of appropriate anesthesia techniques to ensure a safe surgical procedure.

In difficult airway cases like this, management options include tracheostomy or jet ventilation, with the choice depending on factors such as anesthesiologist experience, available equipment, urgency, patient co-morbidities, and the site of airway obstruction. Tracheostomy is invasive, and jet ventilation has been associated with severe complications in laryngeal microsurgery [21]. In this case, sufficient preoxygenation with HFNC provided effective oxygenation and reduced the risk of desaturation during airway manipulation. In addition, retaining the oxygen reserve with HFNC does not require practitioners to have complex skills. Moreover, the use of ORi monitoring facilitated the early detection of hypoxemia, enabling prompt intervention to prevent respiratory complications. To our knowledge, there has not been a clinical report of ORi-monitored non-intubated general anesthesia in pregnant patients with airway difficulty.

HFNC is increasingly being utilized for patients with acute hypoxemic respiratory failure, and its applications are expanding to ensure adequate oxygenation. HFNC has

been shown to improve safety in procedures involving a risk of apnea, such as intubation or bronchoscopy [11]. In this particular case, HFNC demonstrated its potential as a more convenient option for managing difficult intubation, as evidenced by ORi monitoring. In other words, ORi monitoring can guide the appropriate approach to difficult intubation. If the ORi decreased rapidly before surgery, we could have concluded that HFNC was not suitable for the patient, and alternative methods to secure the airway would have been considered. However, in this case, the ORi provided sufficient time to safely remove the vocal mass, confirming the appropriateness of using HFNC. Ultimately, the combination of HFNC and ORi monitoring appeared to have a synergistic effect, enhancing their individual benefits. Additionally, the ORi demonstrated a strong correlation with the decline in the pulse oximeter oxygen saturation ($SpO_2$). In our case, the ORi exhibited a more rapid decrease compared to the $SpO_2$. This indicates that the ORi can serve as a useful guide for tracking the $SpO_2$ decline, allowing for the efficient determination of the need for additional oxygenation.

However, the ORi, which measures oxygen levels based on the finger wavelength, can be influenced by the finger's blood flow, temperature, and carbon dioxide content, impacting the interpretation of ORi monitoring [13]. In situations involving high-dose vasopressors or shock, the accuracy of ORi monitoring decreases. Additionally, it is important to note that the ORi is not equivalent to the arterial oxygen content and cannot replace blood gas tests. HFNC also has certain limitations. It cannot be used in patients with facial trauma, and caution should be exercised in cases with a higher risk of aspiration.

Anesthetic management for pregnant patients with a bilateral vocal cord granuloma necessitates a customized approach to optimize oxygenation and minimize the potential of the airway being compromised. The innovative approach in a pregnant patient with a bilateral vocal cord granuloma offers valuable insights for patient safety and surgical success. This case report emphasizes the challenges encountered, the selected anesthetic management strategy, and the benefits of using HFNC oxygenation with ORi monitoring in this unique scenario. The combination of high-flow nasal cannula oxygenation and ORi monitoring effectively ensured sufficient oxygenation and enabled the early detection of hypoxemia in this case. Further research and more extensive studies are necessary to validate the benefits of this approach in similar cases.

**Author Contributions:** Conceptualization, S.K.O.; methodology, J.E.L., J.G.C. and S.K.O.; writing—original draft preparation, H.S.K. and S.K.O.; writing—review and editing, H.S.K., H.A.L. and S.K.O.; visualization, J.E.L.; supervision, S.K.O. All authors have read and agreed to the published version of the manuscript.

**Funding:** This research received no external funding.

**Institutional Review Board Statement:** Ethical review and approval were waived for this anonymized single-case report.

**Informed Consent Statement:** Written informed consent was provided by the patient, including the use of photographic images.

**Data Availability Statement:** Not applicable.

**Conflicts of Interest:** The authors declare no conflict of interest.

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
