# Peer review of "Anesthetic Management for a Pregnant Patient with Bilateral Vocal Cord Granuloma Using High-Flow Nasal Cannula Oxygenation with Oxygen Reserve Index Monitoring: A Case Report"

_2038-9582, doi:10.3390/std12030015_

Round 1

Reviewer 1 Report

The authors of this study presented a case of successful anesthetic management for pregnant suffering from airway pathology caused by traumatic granuloma. This approach may be a good option for managing granuloma in general, which can certainly help anesthesiologists and otolaryngologists when deciding on the most carefree procedure. Therefore, I prefer such works that describe the procedure in detail and that can have an impact on similar procedures and resolve the dilemmas that clinicians sometimes face. I congratulate the authors, and the questions are as follows: 1. explain the use of stethoids and proton pump inhibitors with regard to the teratogenic effect? 2. why did they do an X-ray scan on a pregnant woman?

Author Response

  1. explain the use of stethoids and proton pump inhibitors with regard to the teratogenic effect?

Author’s Answer: We would like to thank the reviewer for their careful and thorough reading of this manuscript and for the thoughtful comments and constructive suggestions. However, as steroid classified as category C of teratogen and it’s potential of increased risk of cleft palate during 1st trimester, short period of steroid usage was recommended by obstetrician.

We added this in line 100 to 104 as follows with references;

“Initially, oral prednisolone 20mg daily for 7 days, following prednisolone 10mg daily for 7 days with proton pump inhibitor were prescribed for treatment. Although the limited in-formation, evidence suggests that proton pump inhibitors are not teratogenic [14]. How-ever, as steroid classified as category C of teratogen and it’s potential of increased risk of cleft palate during 1st trimester, short period of steroid usage was recommended by obste-trician [15].”

  1. why did they do an X-ray scan on a pregnant woman?

Author’s Answer: Thank you for the question.

The patient had dyspnea, so she underwent chest x-ray in case of possibility of combined pulmonary complication, but with shielding to reduce fetal radiation.

We added this in line 128 to 129 as follows;

“Due to dyspnea, the patient underwent chest x-ray with shielding to reduce fetal radiation [16]….”

Reviewer 2 Report

This is a good manuscript highlight efficient approach for managing vocal cord granuloma in pregnant patient with challenges in the intubation.

Minor revision is necessary as in yellow highlight.

Thank you.

This is a good manuscript highlight efficient approach for managing vocal cord granuloma in pregnant patient with challenges in the intubation.

The English is good already.

Minor revision is necessary as in yellow highlight.

Thank you.

Author Response

Author’s Answer: We would like to thank the reviewer for their careful and thorough reading of this manuscript and for the thoughtful comments and constructive suggestions.

Following your suggestion, arrows were added to Figure 1, (we additionally inserted in Figure 2 for more information), and anatomical structures were displayed in Figure 3 (former Figure 2 before revision).

Spacing and punctuation have also been corrected, thanks to your meticulous point out.

Also, the part that needed to be rephrased was modified as follows.

Line 191-192; (before revision, “….the patient provided reassurance by reporting the absence of any recurring symptoms that 172 were experienced prior to the development of bilateral granuloma.”)  

->  “….the patient showed an absence of any recurring symptoms that existed prior to the development of bilateral granuloma.”

Line 267; (before revision, “……the potential for airway compromise”)

-> “……the potential of airway compromise”